# Comparative Transcriptome Analysis Reveals Epithelial Growth Factor Receptor (EGFR) Pathway and Secreted C-Type Lectins as Essential Drivers of Leg Regeneration in *Periplaneta americana*

**DOI:** 10.3390/insects16090934

**Published:** 2025-09-05

**Authors:** Xiaoxuan Liu, Nan Sun, Xiaojuan Wu, Jiajia Wu, Shuqi Xian, Dayong Wang, Yechun Pei

**Affiliations:** 1Collaborative Innovation Center of Lift and Health, Hainan Province Key Laboratory of One Health, School of Life and Health Sciences, Hainan University, Haikou 570228, China; 22110710000002@hainanu.edu.cn (X.L.); 23220860010008@hainanu.edu.cn (X.W.); wujiajia0224@icloud.com (J.W.); 24220860010084@hainanu.edu.cn (S.X.); 2School of Tropical Agriculture and Forestry, Hainan University, Haikou 570228, China; sunnan@hainanu.edu.cn; 3Hainan International One Health Institute, Hainan University, Haikou 570228, China; 4Laboratory of Biopharmaceuticals and Molecular Pharmacology, School of Pharmaceutical Sciences, Hainan University, Haikou 570228, China

**Keywords:** *Periplaneta americana*, regeneration, RNA-seq, *EGFR*, *Regenectin*

## Abstract

The American cockroach (*Periplaneta americana*) serves as a promising model organism for investigating regeneration due to its remarkable ability to regenerate appendages. This study explores the molecular mechanisms underlying leg regeneration in *P. americana*. We performed amputations and documented the regenerative process using hematoxylin and eosin (H & E) staining. Based on morphological analysis of H & E-stained sections and existing research findings, we examined blastema formation and cell proliferation (~3–7 dpa), followed by morphogenesis and cell differentiation (~7–14 dpa). Through comparative transcriptomic analysis of samples collected at different stages of limb regeneration, we identified numerous differentially expressed genes potentially associated with growth or regenerative processes. Notably, our results imply that the Epidermal Growth Factor Receptor (EGFR) signaling pathway and the C-type lectin (Regenectin) may play critical roles in regulating leg regeneration in *P. americana*. These findings offer a theoretical foundation for understanding regeneration mechanisms in cockroaches and provide a basis for further exploration of the molecular regulatory networks involved.

## 1. Introduction

“If there were no regeneration, there would be no life.” [1]. Regeneration describes the process whereby an organism can reconstruct partial or complete tissue structures at injury sites, following damage to its tissues or organs. Leg regeneration occurs in certain animal phyla [2], with regenerative capabilities varying significantly between species. Planarians demonstrate remarkable regenerative abilities, capable of regenerating entire organisms through neoblast proliferation, representing an exceptional model system extensively studied in regenerative biology [3]. Zebrafish (*Danio rerio*), tadpoles, and salamanders exhibit robust appendage regeneration, particularly in caudal fin or limb restoration [4,5]. Conversely, *Mus musculus* and humans display limited regenerative capabilities, primarily restricted to specific organs or tissues [6]. Among insects and arthropods, regenerative capacity varies substantially, with holometabolous insects like *Drosophila melanogaster* showing relatively limited regenerative abilities. Hemimetabolous insects, particularly cockroaches, serve as valuable models in regeneration research due to their exceptional capacity for limb regeneration, demonstrating the most pronounced appendage regenerative capabilities among arthropods [7].

Insect limb regeneration, analogous to appendage regeneration in other arthropods [8,9], comprises three distinct developmental phases: wound epithelium formation, blastema generation, and morphogenetic patterning [10]. Following leg amputation, the initial phase involves wound closure, where haemocytes rapidly adhere to the wound surface. During the second phase, a melanized scab forms at the site, sealed by hemolymph clotting. Notably, this phase exhibits minimal cell proliferation [8]. Research indicates that the wound repair primarily relies on migratory cellular mechanisms rather than mitotic activity [11]. Blastema formation represents a crucial stage in tissue regeneration. The process heavily depends on blastema cells proliferation, typically involving dedifferentiated cells near the wound epidermis, which subsequently proliferate to restore the missing tissue’s function [7]. Consequently, most regeneration-capable animals form a blastema at the injury site [7,12], whereas mammals cannot regenerate limbs post-amputation due to scar tissue formation instead of blastema development. Several essential genes and signaling pathways participate in various regeneration stages, including fibroblast growth factor (FGF) [13], insulin-like growth factor-I [14], epithelial growth factor receptor (EGFR) [15,16], Janus kinase-signal transducer and activator of transcription (JAK/STAT) [17,18], Hedgehog (Hh) [19] and Decapentaplagic (Dpp) [20].

*Periplaneta americana* exhibits remarkable survival capabilities. This resilience stems from its exceptional tissue repair and limb regeneration abilities. Particularly notable is its appendage regeneration capacity: nymphs can regenerate complete, functional appendages after a single molting cycle following damage or loss. Additionally, *Periplaneta americana* serve as a raw material in traditional Chinese medicine. Research demonstrates that oligopeptides isolated from *P. americana* enhance wound healing [21]. Current research on *P. americana* regeneration research primarily examines blastema formation mechanisms [22,23]. The second Keystone symposium expanded regeneration conceptual frameworks by emphasizing growth mechanisms [24]. Understanding the relationship between regenerative and homeostatic growth control/maintenance remains essential, as maintaining homeostatic growth control is crucial during wound healing and regeneration [25].

This study documents the regeneration process and performs comparative transcriptomic analyses on samples from different regeneration stages of *P. americana*, with particular attention on growth- or regeneration-related genes. These findings advance our understanding of regeneration mechanisms.

Our analysis revealed the involvement of the EGFR signaling pathway and C-type lectins in blastema formation & proliferation and/or morphogenesis & differentiation. These discoveries contribute to both *P. americana* resource development and enhanced understanding of regeneration’s molecular mechanisms.

## 2. Materials and Methods

### 2.1. Experimental Animals

*P. americana* specimens were sourced from an American cockroach breeding farm in Chizhou city, Anhui province, China. The colony was maintained under controlled conditions (28 °C, 70–80% relative humidity) in ventilated plastic boxes, with animals were provided with lab mice diet and purified water. To synchronize animal cohorts, oothecae were hatched uniformly with molt numbers recorded. ~8th instar nymphs were selected from the colony and housed in separate plastic boxes, where they were supplied with lab mice diet and purified water.

### 2.2. Histological Analysis

To investigate the morphological changes during appendage regeneration, Coxa samples were collected at 0 h, 3 dpa, 7 dpa, and 14 dpa post-amputation (dpa), with three biological replicates per group. The samples were fixed in paraformaldehyde solution for 24 h, followed by dehydration through a graded ethanol series (70%, 85%, 95%, and 100%). Subsequently, tissues were cleared in xylene and paraffin-embedded. Sections (Transverse Plane, ~6 µm) were cut from the paraffin blocks and stained with hematoxylin and eosin (H & E). The stained sections were mounted with resinous medium, observed under a bright-field microscope, and photographed (Leica DM 500, Wetzlar, Germany).

### 2.3. Collection of Biological Material

The experimental cockroaches were anesthetized with ice for sample collection. For RNA-seq, a hind-leg was amputated at the first day post molting. The distal tarsus, tibia and femur of the hind-leg were initially removed at the junction of the trochanter and femur, leaving the trochanter and coxa intact. The entire coxa and trochanter were harvested at both 7 and 14 dpa in the experimental groups. For the control group, the whole coxa and trochanter were collected immediately (0 dpa) post amputation. Each group comprised 15 individuals pooled to constitute a single composite sample, with three such replicates generated per experimental group.

### 2.4. RNA Extraction and Sequencing

Total RNA was extracted from tissues using TRIzol^®^ Reagent (Thermo Fisher Scientific, Waltham, MA, USA) according to the manufacturer’s protocol. RNA integrity and purity were evaluated using an Agilent 5300 Bioanalyzer (Agilent Technologies, Santa Clara, CA, USA) and quantified using a NanoDrop ND-2000 spectrophotometer (Thermo Fisher Scientific, Waltham, MA, USA). Only RNA samples meeting quality criteria (OD260/280 = 1.8~2.2, OD260/230 ≥ 2.0, RQN ≥ 6.5, 28S:18S ≥ 1.0, >1 μg) were selected for sequencing library preparation. Transcriptome sequencing was performed on an Illumina platform by Majorbio Technologies Company (Shanghai, China). The ligation library was constructed using 1 µg of total RNA. Polyadenylated mRNA was enriched through oligo (dT) bead-based poly (A) selection and subsequently fragmented. Double-stranded cDNA synthesis was conducted using a SuperScript™ Double-Stranded cDNA Synthesis Kit (Thermo Fisher Scientific, Waltham, MA, USA) primed with random hexamers (Illumina). Libraries were prepared following the Illumina protocol involving end-repair, phosphorylation, and ‘A’ base addition. Target cDNA fragments (~300 bp) were size-selected using 2% Low Range Ultra Agarose gel electrophoresis. The size-selected fragments underwent PCR amplification with Phusion High-Fidelity DNA Polymerase (NEB) (Thermo Fisher Scientific, Waltham, MA, USA) for 15 cycles. Library quantification was performed using Qubit 4.0, and paired-end sequencing (2 × 150 bp) was conducted on the NovaSeq X Plus platform (Illumina, San Diego, CA, USA).

### 2.5. Quality Control and Read Mapping

Raw paired-end reads underwent quality trimming and control using fastp (Version 0.19.5) [26] with default parameters. Clean reads were then aligned to the reference genome in a strand-specific manner using HISAT2 (Version 2.1.0) [27]. Transcript assembly for each sample was performed using StringTie (Version 2.1.2) [28] in a reference-guided approach.

### 2.6. Differential Expression Analysis and Functional Enrichment

Transcript abundance was quantified using RSEM and normalized to transcripts per million (TPM). Differential expression analysis between sample groups was performed using differentially expressed genes (DEGs) with |log2FC| ≥ 1 and FDR < 0.05 (DESeq2) [29]. Genes with |log2FC| ≥ 1 and FDR < 0.05 (DESeq2) were classified as significantly DEGs. To understand the functional implications of the identified DEGs, enrichment analyses for Gene Ontology (GO) terms and Kyoto Encyclopedia of Genes and Genomes (KEGG) pathways were conducted. The GO database provides functional annotation of screened genes, encompassing biological processes (BP), cellular components (CC), and molecular functions (MF). The KEGG database categorizes genes within a gene set based on their associated biochemical pathways or molecular functions, encompassing environmental information processing (EIP), metabolism (M), cellular processes (CP) and organismal systems (OS). Statistical significance was determined using a Bonferroni-corrected *p*-value < 0.05 relative to the whole transcriptome background. GO enrichment utilized Goatools (Version 0.6.5), while KEGG pathway analysis employed SciPy (Version 1.7.1).

### 2.7. Real-Time Quantitative PCR

For gene expression analyses, 10-fold diluted cDNA (synthesized from the same RNA batch as used for RNA-seq library preparation) served as templates for qRT-PCR. Reactions were performed in triplicate using ChamQ Universal SYBR qPCR Master Mix (Vazyme, Nanjing, China). The relative expression levels were calculated using the 2^−ΔΔCt^ method and there were three biological replicates and three technical replicates for each sample. The primers used for qRT-PCR are shown in Appendix A.

### 2.8. Double-Stranded RNA (dsRNA) Treatment

dsRNA was synthesized using a T7 RiboMAX Express RNA interference (RNAi) System (Promega, Madison, WI, USA, P1700). Following purification, a dsRNA solution of 2 μg/μL was microinjected into the coxa of each nymph, delivering 2 μg per individual. The dsRNA injections were administered every 7 days to maintain RNAi efficiency throughout the regeneration process. The dsRNA-synthesizing primer sequences are detailed in Appendix A.

## 3. Results and Discussion

### 3.1. Morphological Profiles and Histological Analysis of Regeneration Legs

Research demonstrates that the trochanter plays a crucial role in leg regeneration in *Periplaneta americana* [20]. Amputations performed at both the trochanter and femoral joints enable nymphs to regenerate fully functional limbs with complete morphology following a single molt cycle (Figure 1A). To examine regeneration dynamics, histochemical analysis through hematoxylin and eosin (H & E) staining was conducted at 0, 3, 7, and 14 dpa. At 0 dpa, cells near the wound surface of the appendage exhibited a loose internal arrangement (Figure 1B). Between 0 dpa and 3 dpa, numerous cells migrated toward the wound area, with blastema formation occurring at the wounded trochanter (~3 dpa) (Figure 1C), aligning with observations in 4th-instar nymphs [23]. The blastema subsequently elongates, accompanied by the formation of the original shape of the new leg (~7 dpa) (Figure 1D). During this phase, regenerating tissues continue developing and folding inside the coxa until expansion after molting (~7–14 dpa) (Figure 1E). Based on morphological analysis of H & E-stained sections and existing research findings, we examined blastema formation and cell proliferation (~3–7 dpa), followed by morphogenesis and cell differentiation (~7–14 dpa) in 8th-instar nymphs. Compared to 4th-instar nymphs, 8th-instar nymphs exhibited demonstrated extended phases of cell proliferation and morphogenesis, indicating a slower developmental progression. This observation establishes a robust foundation for our subsequent transcriptomic investigations.

### 3.2. Analyses of Sequencing Quality

The transcriptomic analysis revealed high-quality sequencing data. Each insect sample generated a minimum of 42.37 million clean data reads. The data quality was validated by several metrics: a minimum of 96.34% of clean data achieved a quality score of Q30, mapping ratios ranged from 84.01% to 89.5%, unique mapping rates spanned from 81.11% to 86.65% and the multiple-mapping rates varied from 2.6% to 2.94%. These parameters confirmed the completeness and quality of the assembly (Appendix A). The analysis yielded 58,448 transcripts, which were subsequently aligned against multiple databases (NR, Swiss-Prot, Pfam, EggNOG, GO, and KEGG) for comprehensive functional annotation through sequence homology and domain characterization (Figure 2). The analysis revealed that 53,814 transcripts (92.07%) matched with insect protein sequences in the Non-Redundant Protein Sequence Database (NR) (Appendix A). Principal Component Analysis (PCA) demonstrated that regenerative stages (0 dpa, 7 dpa, 14 dpa) accounted for the majority of variance (PC1), while the distinct clustering of samples along PC1 indicated a time-dependent progression in gene expression patterns (Figure 3).

### 3.3. Differential Expression of Genes

A previous transcriptional study identified several essential pathways in *Periplaneta americana* leg regeneration, including JAK/STAT, Dpp, Notch, Wnt/Wingless (Wg), and Hedgehog [23], This study reports significantly differentially expressed growth-related genes that demonstrated statistically significant changes in expression between three points, meeting the criteria *p*-adjust < 0.05, |log2FC| ≥ 1. Between 7 dpa and 0 dpa, 2343 DEGs were identified, comprising 798 upregulated and 1545 downregulated genes (Figure 4, Appendix A). Among upregulated growth- or regeneration-related genes in regenerating legs were extracellular matrix components (different collagen isoforms and *tektin-2* (*tekt2*)), insulin-like growth factor (*igf*), and secreted C-type lectin gene-8 (*sClec-8*). Downregulated genes at the earliest post-amputation time point (7 dpa) were associated with metabolism (*glucose dehydrogenase* (*gdh*), *Lipase* and *glucose-6-phosphate dehydrogenease* (*g6pd*)) and immune-related genes (*Toll-like recrptor 13* (*tlr13*), *defensin* (*def*), *drosomycin-like defensin* (*dld*), *Polyphenoloxidases2* (*ppo2*), *cytochrome P450* (*p450*), *The peptidoglycan recognition protein* (*pgrp*) and *sClecs*).

Between 14 dpa and 0 dpa groups, 2963 DEGs were identified, with 1348 upregulated and 1615 downregulated genes (Figure 4, Appendix A). Numerous genes were upregulated, including *cuticle* and *Pro-resilin*, extracellular matrix components (different collagen isoforms, *laminin subunit gamma-1* (*LAMC1*), *Integrin beta* (*itgb*) and *matrix metalloproteinase* (*mmp*)), cell proliferation related genes (*igf*, *protein phosphatase 1* (*pp1*) and *Cyclin D2* (*ccnd2*)), metabolism (*gdh*), and secreted C-type lectin genes (*regenectin*, *sClec-1 and sClec-8*). Immune-related genes (different secreted C-type lectin, *tlr13*, *tlr4*, *def*, *dld*, *ppo2*, *p450* and *pgrp*) maintained their downregulated status.

Between 14 dpa and 7 dpa groups, 3135 DEGs were identified, comprising 1760 upregulated and 1375 downregulated genes (Figure 4, Appendix A). The analysis revealed further upregulation of extracellular matrix and cell proliferation related genes, as well as secreted C-type lectin genes (*Regenectin*, *sClec-1 and sClec-8*) in the 14 dpa group compared to the 7 dpa group. Conversely, immune-related genes exhibited further downregulation in the 14 dpa group relative to the 7 dpa group.

A heatmap of differentially regulated growth-related genes is shown in Figure 5A. Extracellular matrix (ECM) proteins modulate cellular processes including proliferation, migration, and differentiation. Furthermore, they provide essential structural support for regenerating legs [30]. Multiple ECM genes showed upregulated expression during regeneration (Figure 5B). For instance, collagen functions as a biomaterial providing structural support for muscle cells during regeneration [31], and is considered a key component in most formulations developed for wound healing applications [32]. Insect Pro-resilin, an elastic protein with rubber-like properties [33], typically cross-links with cuticles or other functional components [34]. Matrix metalloproteinase important in regulating ECM homeostasis [35], was found to be dispensable for leg regeneration [23]. The results indicated that *EGFR* was upregulated at 14 dpa. In skin regeneration, the EGFR-COL17A1 axis-mediated keratinocyte stem cell motility drives epidermal regeneration [36]. In *Gryllus bimaculatus*, EGFR signaling pathway regulates positional specification along the proximodistal axis during distal leg development in insects [15]. Thus, EGFR signaling pathway may play a significant role in the regeneration of *P. americana.* Notably, several secreted C-type lectin genes showed upregulated expression, which function in recognizing oligosaccharides attached to proteins in the ECM [37] and facilitate cell–cell adhesion, immune responses, and intracellular signal transduction [38]. Research has demonstrated that CLEC3A stimulates the Phosphoinositide 3-kinase (PI3K)/protein kinase B (AKT) pathway to accelerate cell proliferation [39]. The upregulated expression of C-type lectin and other ECM components during regeneration suggest their significant contribution to leg regeneration in *Periplaneta americana*.

Additionally, immune and metabolism-related genes, including *peptidoglycan recognition proteins* (*pgrps*), *drosomycin-like defensin* (*dld*), *defensin* (*def*), *cytochrome P450* (*p450*), *secreted C-type lectin* (*sClec*), and *Toll-like receptor* (*tlr*) exhibited downregulated expression (Figure 5C,D). The Growth and Regeneration Keystone meeting highlighted that immune response likely modulates development, homeostasis, wound healing, and regeneration [25]. In mammals, various immune cells mediate tissue repair and regeneration following injury, orchestrating both inflammatory responses and regenerative processes [40]. These findings suggest that *P. americana* may redirect energy and resources toward tissue reconstruction through immune signaling inhibition during regeneration.

### 3.4. Functional Annotation and Pathway Assignment DEGs

Gene Ontology and KEGG analyses were conducted to identify the biological processes and pathways involved in growth regulation of *P. americana*.

Based on Gene Ontology annotation, DEGs were classified into five functional categories (Figure 5A,B, Appendix A), such as proteasome complex, endopeptidase complex, peptidase complex and proteolysis involved in protein catabolic process between 0 dpa and 7 dpa upregulated genes and between 14 dpa and 7 dpa downregulated genes, membrane, cellular anatomical entity and cellular component between 7 dpa and 0 dpa downregulated genes and between 14 dpa and 7 dpa upregulated genes, structural constituent of cuticle, structural molecule activity and chitin binding between 14 dpa and 0 dpa upregulated and oxidoreductase activity, catalytic activity between 14 dpa and 0 dpa downregulated.

KEGG pathway enrichment analysis revealed significant associations with multiple biological pathways in both groups (Figure 6B, Appendix A). The upregulated genes between 7 dpa and 0 dpa were associated with Proteasome, Ubiquitin mediated proteolysis and Protein processing in endoplasmic reticulum pathways. Between 14 dpa and 7 dpa, upregulated genes were associated with Amino sugar and nucleotide sugar metabolism, Hippo signaling pathway, ECM-receptor interaction and PI3K-Akt signaling pathway. The upregulated genes between 14 dpa and 0 dpa were associated with Remin-angiotensin system, Protein digestion and absorption and Adherens junction pathways.

In *Holothuria glaberrima*, proteasomes play an essential role during intestinal regeneration [41]. The present study identified upregulated genes primarily related to proteolysis within 7 dpa, including Proteasome, Ubiquitin mediated proteolysis, proteolysis involved in protein catabolic process and ubiquitin-dependent protein catabolic process, while upregulated genes at 14 dpa were predominantly structure-related. These findings suggested that during the early regenerative phase of *P. americana*, the ubiquitin-proteasome system (UPS) facilitates rapid degradation of damaged or redundant proteins, providing essential amino acids and energy substrates for subsequent cellular remodeling processes [42,43]. The Hippo pathway regulates regeneration by orchestrating cellular proliferation, survival, differentiation, and mechanics [44], a function validated in *P. americana* [23] and *Danio rerio* [45]. ECM-receptor interaction modulates signaling pathways including PI3K and mitogen-activated protein kinase (MAPK), which are crucial in myelin formation and regeneration [46]. Studies have demonstrated that ECM components participate in regenerative processes [47,48,49]. Additionally, PI3K-Akt signaling, regulated by EGFR [50] and IGF signaling pathway [51], can enhance corneal cell migration and proliferation through PI3K-Akt pathway activation.

### 3.5. Validation of Differentially Expressed Transcripts by qRT-PCR

To experimentally validate the transcriptome sequencing results, ten growth-related DEGs were selected from the coxa of *P. americana* for qPCR analysis. (*Actin* was used for normalization). These DEGs included *secreted C-type lectin-1* (*sClec-1*), *sClec-5*, *sCLec-7*, *sCLec-8*, *Regenectin*, *EGFR*, *itgb*, *mmp*, *pointed* (*pnt*) and *tlr*. The expression patterns obtained through qRT-PCR for each DEG aligned with the RNA-seq results (Figure 7A,B). The findings revealed that *EGFR* exhibited upregulated at 14 dpa, although expression of *EGFR* downstream signaling proteins, including *PI3K*, *Extracellular regulated protein kinases* (*ERK*), *JAK*, and *SRC proto-oncogene*, *non-receptor tyrosine kinase* (*Src*) [52], were not observed. The downstream transcription factor *pnt* showed upregulation. While previous studies have established that *pnt* functions in eye development in *D. melanogaster* [53] and juvenile hormone biosynthesis in *P. americana* [54], its regenerative role remains undefined. These observations suggest that *EGFR* may serve a crucial function primarily during morphogenesis, rather than in blastema formation and proliferation, during leg regeneration in *P. americana*.

Notably, multiple differentially expressed C-type lectins were identified during the regeneration process of *P. americana*. Although current research on C-type lectins predominantly examines their immunomodulatory functions [55,56] and development [57,58], few studies have investigated their developmental or regenerative roles. For example, a specific C-type lectin (*Regenectin*) was found to be abundantly expressed and secreted into the regenerating leg during *P. americana* leg regeneration [59]. Additionally, CLEC3A has demonstrated the ability to activate the PI3K-AKT signaling pathway, thereby promoting cellular proliferation [39]. These findings indicate that secreted C-type Lectins participate in leg regeneration in *P. americana*.

### 3.6. Functional Analysis of EGFR and Regenectin During Leg Regeneration

EGFR serves a crucial role in development and tissue homeostasis, specifically facilitating tissue repair and regeneration through activation of signal transduction cascades [52], including extracellular signal-regulated kinase (ERK)-MAPK), PI3K-AKT, SRC and JAK/STAT pathways [60]. In vitro, in vivo, and bioinformatic analyses demonstrated that the regenerating (REG) family member REG3A, a secreted C-type lectin, promotes PDAC cell line growth through direct binding to the extracellular domain of EGFR [61]. This study identified multiple secreted C-type lectins showing upregulated expression, such as *Regenectin*, *sClec1* and *sClec8*. In situ hybridization revealed *Regenectin* mRNA expression exclusively within the newly formed epidermis of regenerating limbs. No expression was detected in developing muscle cells residing inside the regenerating epidermal saccules [59]. However, the mechanism of action underlying Regenectin remains unclear.

To preliminarily investigate the roles of *EGFR* and *Regenectin* in regeneration and their potential functional relationship, RNAi was conducted to knockdown the expression of *EGFR* and *Regenectin* in nymphs. The silencing efficiency of *EGFR* and *Regenectin* were measured on the fourteenth day, and the mRNA levels of *EGFR* and *Regenectin* were significantly reduced compared with control (Figure 8A,B).

Compared to the ds-*CK* control group (Figure 8C), virtually no new tissue regeneration was observed in the ds-*EGFR* group (Figure 8D). This finding differs from previous reports [23], possibly because 8th-instar cockroaches require a longer period to achieve leg tissue regeneration, and *EGFR* interference disrupted blastema formation and/or morphogenesis. This also suggests that EGFR signaling pathway affects leg regeneration in *P. americana*.

A preliminary investigation of Regenectin’s mechanism of action on leg regeneration was performed. Regenerated legs in the ds-*Regenectin* group exhibited shortening and malformation (Figure 8E), indicating that Regenectin is essential for normal leg regeneration in *P. americana*. The regenerative length was also measured after RNAi treatments of genes (Figure 8F). Therefore, initial assessment focused on EGFR signaling pathway gene expression following ds-*Regenectin* administration. The results revealed that the expression levels of genes (*Spitz* and *EGFR*) were all significantly reduced in the ds-*Regenectin* group (Figure 8G). In vitro analysis demonstrated that Collectin-11, a soluble C-type lectin, activate EGFR signaling pathway and directly stimulate murine melanoma cell proliferation [62]. Notably, the expression of *Regenectin* and another C-type lectin gene (*sClec1*) were upregulated after knockdown of *EGFR* (Figure 8H), suggesting that *Regenectin* may mediate leg regeneration of *P. americana* by regulating the EGFR signaling pathway. However, this study only analyzed the transcription level of *EGFR*, and we did not determine the spatial expression pattern of *EGFR* in regenerating legs or EGFR signaling pathway activity. Therefore, the potential regulatory relationship between limb regeneration and EGFR signaling pathway requires further investigation.

## 4. Conclusions

This study investigated the morphological changes during leg regeneration of *Periplaneta americana*. Through comparative transcriptome analysis, we identified 86 DEGs associated with growth or regeneration processes, providing valuable data for future research into the molecular mechanisms of leg regeneration in *P. americana*. The findings demonstrate that epidermal growth factor receptor (EGFR) and secreted C-type lectin (Regenectin) regulate leg regeneration. Additionally, Regenectin may modulate the expression of *EGFR* and its ligand *Spitz*. The dynamic expression patterns and regulatory interactions between *EGFR* and *Regenectin* during leg regeneration warrant further investigation. These findings advance our understanding of the molecular mechanisms underlying leg regeneration in *P. americana*.

## Figures and Tables

**Figure 1 insects-16-00934-f001:**
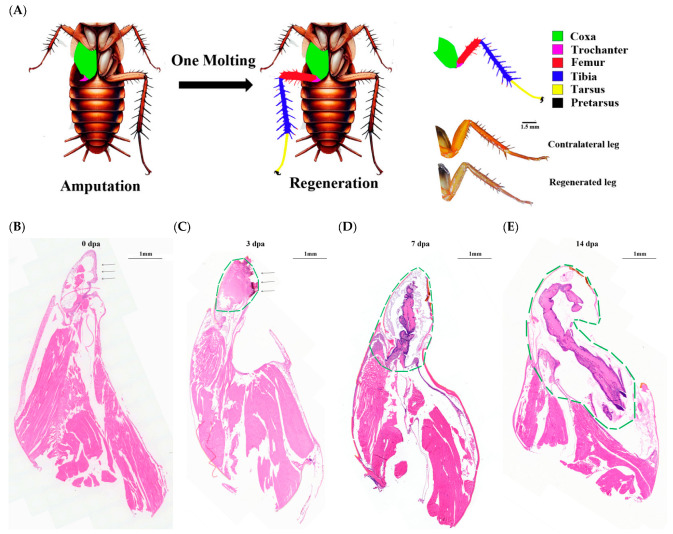
Morphological and HE staining of regenerating legs under transection. (**A**): Schematic Diagram of the Regeneration Process and Regeneration Legs. Six different colors represent different segments of legs. (**B**): The histological images at 0 days after amputation, (**C**): 3 dpa, (**D**): 7 dpa, (**E**): 14 dpa. The arrows indicate wound sites and green dotted lines cover the regenerative regions. Scale bars are labeled for microscopic images.

**Figure 2 insects-16-00934-f002:**
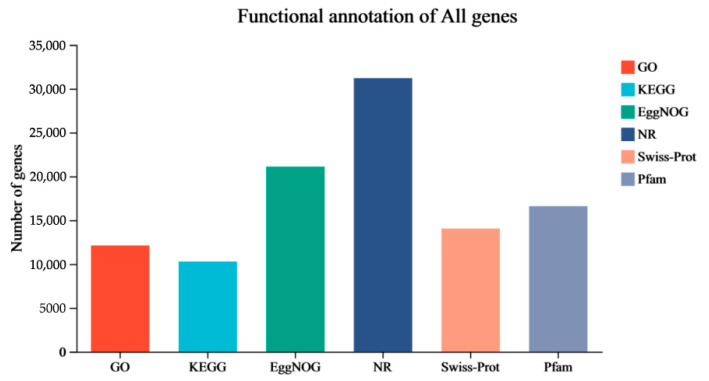
Distribution of transcripts according to their BLAST (Version 2.9.0) results against the respective databases.

**Figure 3 insects-16-00934-f003:**
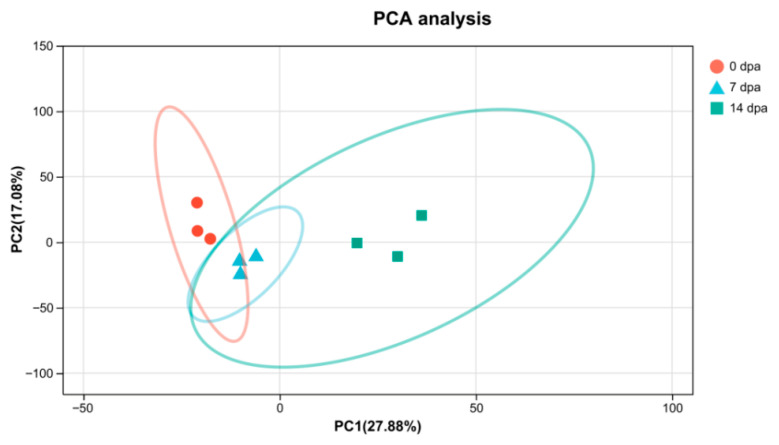
The profiles of genes at three time points during leg regeneration analyzed by the PCA.

**Figure 4 insects-16-00934-f004:**
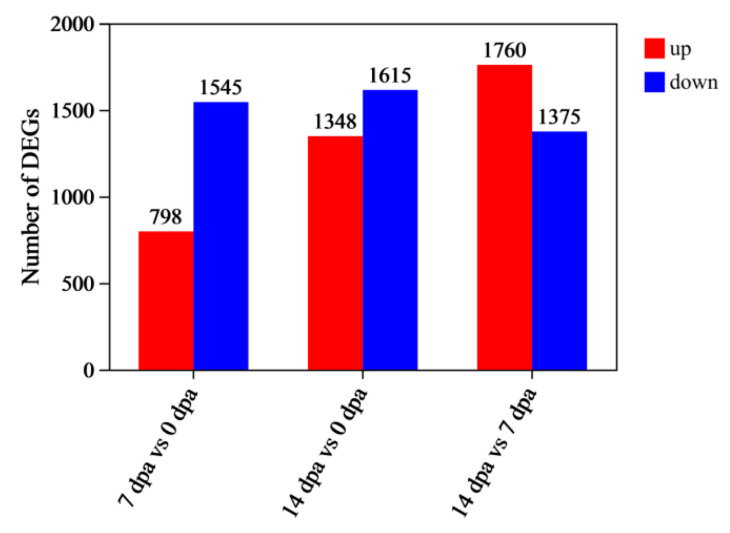
Statistics of differentially expressed genes (DEGs).

**Figure 5 insects-16-00934-f005:**
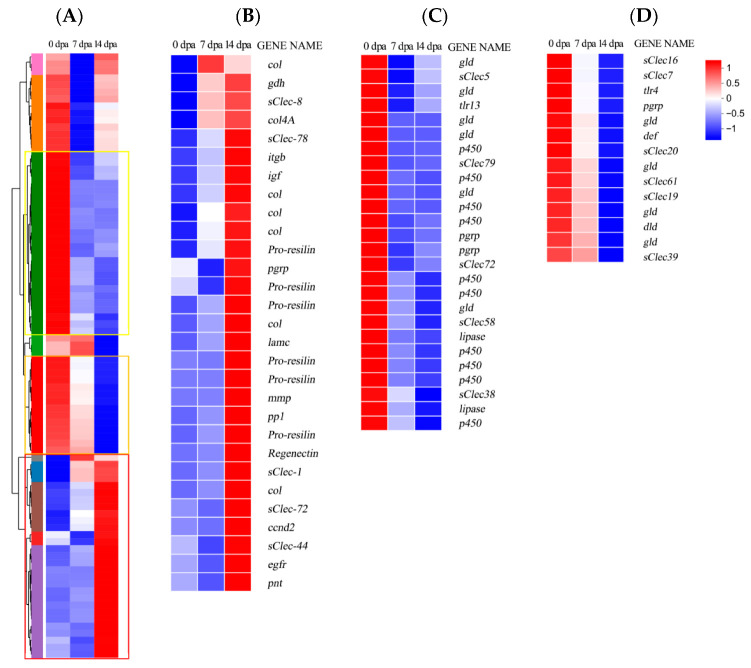
Heatmap of differentially expressed genes during leg regeneration. (**A**): Growth-related DEGs with *p*-adjust < 0.05 and |log2FC| ≥ 1 were analyzed (*n* = 86). Two distinct clusters were identified: high expression (red clusters) and low expression (blue cluster), dpa: days post-amputation. Red bars indicate upregulated genes; blue bars indicate downregulated genes. (**B**): A group of genes exhibiting upregulation at 7 dpa and 14 dpa (red square in (**A**)). (**C**): A group of genes downregulated at 7 dpa (yellow square in (**A**)). (**D**): A group of genes downregulated at 14 dpa (orange square in (**A**)).

**Figure 6 insects-16-00934-f006:**
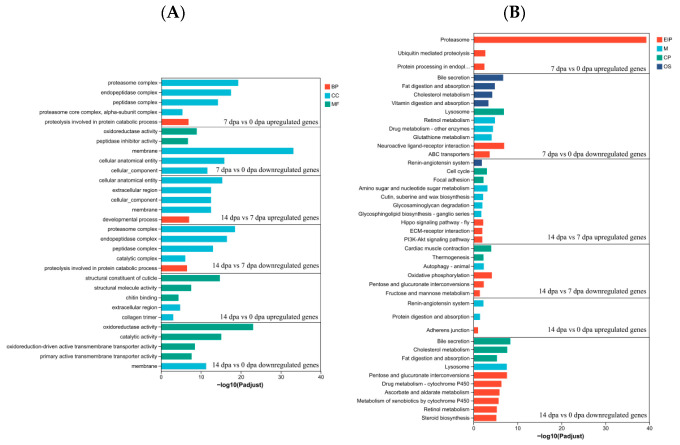
Bar chart visualization for multi-gene set enrichment analysis based on GO annotation (**A**) and KEGG pathways (**B**). A higher −log10 (Padjust) value indicates greater significance of enrichment for the corresponding GO and KEGG term.

**Figure 7 insects-16-00934-f007:**
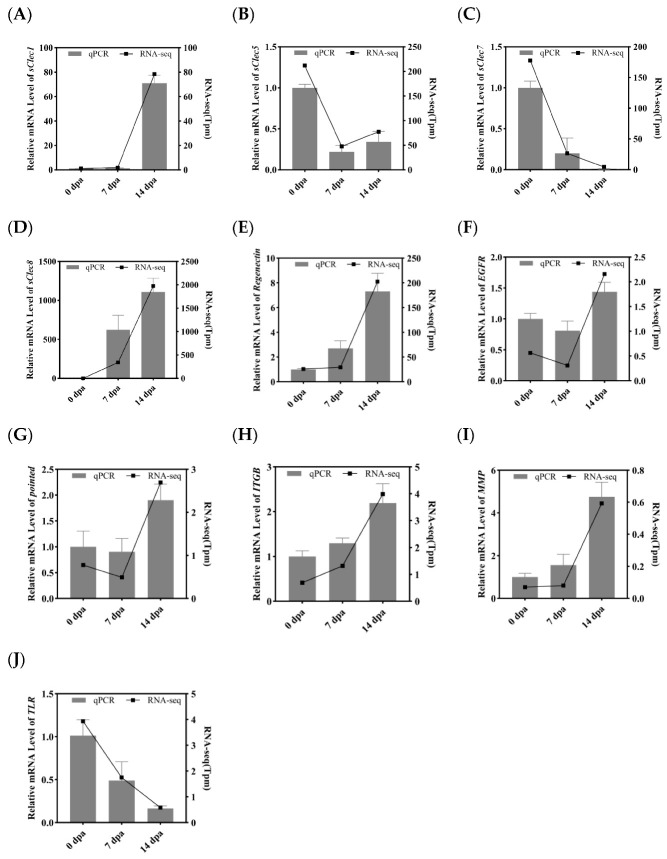
DEGs associated with regeneration were determined by RNA-seq and qRT-PCR. (**A**) Relative mRNA level of *secreted C-type lectin 1* (*sClec1*); (**B**) Relative mRNA level of *sClec5*; (**C**) Relative mRNA level of *sClec7*; (**D**) Relative mRNA level of *sClec8*; (**E**) Relative mRNA level of *Regenectin*; (**F**) Relative mRNA level of *epidermal growth factor receptor* (*EGFR*); (**G**) Relative mRNA level of *pointed*; (**H**) Relative mRNA level of *Integrin beta* (*ITGB*); (**I**) Relative mRNA level of *matrix metalloproteinase* (*MMP*); (**J**) Relative mRNA level of *Toll-like receptor* (*TLR*). Bar charts (left *y*-axis) represent qRT-PCR results, while line graphs (right *y*-axis) indicate RNA-seq results. Bar values represent mean ± SE of three independent measurements.

**Figure 8 insects-16-00934-f008:**
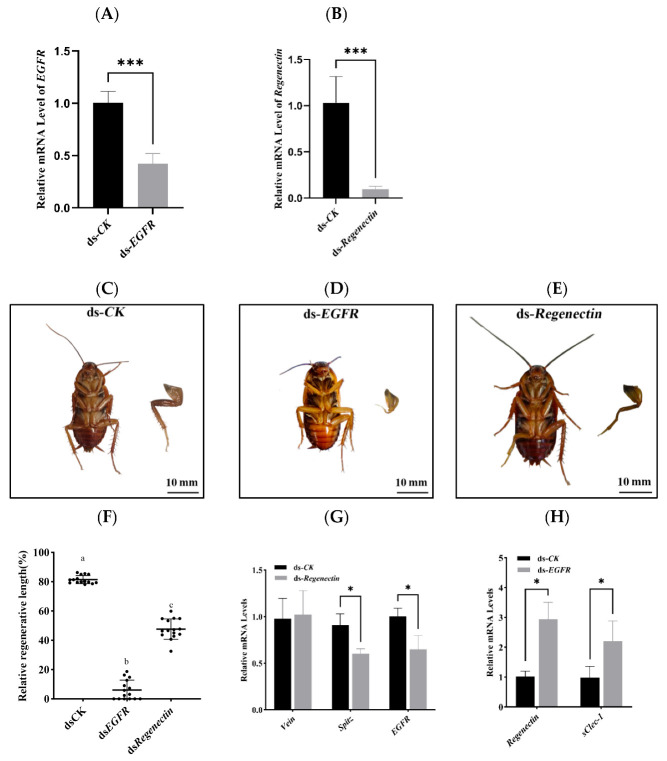
Leg regeneration phenotype of *P. americana*. (**A**). The silencing efficiency of ds-*EGFR* in 14 dpa. (**B**). The silencing efficiency of ds-*Regenectin* in 14 dpa. (**C**–**E**). RNAi resulted in significantly impeded regeneration. (**F**). Relative regenerative length of regenerated legs in dsCK, ds*EGFR* and ds*PaSCLec* groups. The relative regenerative lengths were calculated by dividing the length of the regenerated legs by the length of the contralateral legs. (**G**). The relative mRNA level of *EGFR* signaling pathway genes following *ds-Regenectin* knockdown in 14 dpa. (**H**). The mRNA level of Regenectin following ds-*EGFR* knockdown in 14 dpa. Asterisk indicates significant differences compared with values of the control (Student’s *t* test, * *p* < 0.05, *** *p* < 0.001). Different letters on the error bar indicate statistically significant differences at *p* < 0.05 level (ANOVA in association with Tukey’s HSD test). Each group including 15 larvae was injected into dsRNA.

## Data Availability

RNA-seq data are available in the Sequence Read Archive (SRA) database under accession numbers PRJNA1274893.

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
