# Peer review of "Comparative Transcriptome Analysis Reveals Epithelial Growth Factor Receptor (EGFR) Pathway and Secreted C-Type Lectins as Essential Drivers of Leg Regeneration in Periplaneta americana"

_insects, 2025, doi:10.3390/insects16090934_

Round 1
Reviewer 1 Report
Comments and Suggestions for Authors
The manuscript "Comparative Transcriptome Analysis Reveals EGFR Pathway and Secreted C-type Lectins as Essential Drivers of Leg Regeneration in Periplaneta Americana" is an interesting study that analyzes transcriptomic data of leg regeneration at different times.
Although the work is interesting, the data is not properly presented, the results are barely described, there is no clear discussion, and the conclusion is not a conclusion. The work has interesting information that needs to be exploited to get published, and the authors have to address major concerns, just to give some examples:
-In the introduction, the authors mention traditional medicine. However, there is no scientific evidence supporting the claim that raw or processed cockroach products promote regeneration. The manuscript appears to promote this unverified concept, which borders on pseudoscience and should be critically addressed or removed.
– The manuscript includes a meta-analysis using several tools such as KEGG, GO, and others. However, the rationale for using these tools is not provided, nor are any of them explained. Despite performing such a comprehensive analysis, the authors only highlight 22 "interesting" genes without discussing the remaining data. Given that the transcriptome includes over 50,000 genes. The authors need to provide a more thorough analysis and discussion of the broader findings.
-What are Clec genes, and why are they important?
-In Figure 7, Did was evaluated by qPCR or Digital PCR? It is not clear in lines 276-277 what it means.
-Figure 8C-E, the Figure legend is not clear.
-All the figures refer to elements "as indicated by the arrows," yet no arrows are actually present in the images. Additionally, the figure legends are poorly described and lack sufficient detail. They should be revised to include clear, accurate, and informative explanations that help the reader interpret the figures effectively.
-Tukey's test is not the correct test to compare two groups.
-Why did the authors compare the 0 vs 7 and 7-14? Why didn´t they compare 0 vs 7, and 0 vs 14 dpa, since 0 is the reference?
-Results have to explain why and how they did the experiments.
-Discussion has to support these results with literature.
-The conclusion is a summary; there is no conclusion in the current work.
Comments on the Quality of English LanguageThe manuscript needs to be reviewed by a native English editor.
It contains several mistakes, for example:
-Lines 276–277: It says "of the digital expression date." Should this be "data" instead?
-Some information in the supplementary data is written in Chinese.
Author Response
Comments 1: In the introduction, the authors mention traditional medicine. However, there is no scientific evidence supporting the claim that raw or processed cockroach products promote regeneration. The manuscript appears to promote this unverified concept, which borders on pseudoscience and should be critically addressed or removed.
Response 1: We agree with the reviewer’s assessment. We have deleted this description since this claim lacked supporting evidence and could mislead readers, it was removed from the manuscript.[mention specific location: e.g., Page 2, Section 1, Line 81]
Comments 2:The manuscript includes a meta-analysis using several tools such as KEGG, GO, and others. However, the rationale for using these tools is not provided, nor are any of them explained. Despite performing such a comprehensive analysis, the authors only highlight 22 "interesting" genes without discussing the remaining data. Given that the transcriptome includes over 50,000 genes. The authors need to provide a more thorough analysis and discussion of the broader findings.
Response 2: Thank you for pointing this out. I agree with this comment. Therefore, I have explained the principles underlying these tools.[mention specific location: e.g., Page 4, Section 2.6, Lines 143-146] In the Results and Discussion, I conducted a broader screening of growth- or regeneration-related differentially expressed genes. I visualized the expression levels of these genes across various stages using heatmap. Additionally, I re-analyzed and discussed these findings.[mention specific location: e.g., Page 6, Section 3.3, Lines 210-279]
Comments 3:What are Clec genes, and why are they important?
Response 3: We thank the reviewer for this important question regarding Clec genes. Clec genes encode a large family of proteins called C-type lectins or CTLs. sClec genes encode a distinct subgroup of soluble C-type lectins. Unlike the membrane bound Clec receptors, these proteins are secreted into body fluids where they function as pattern recognition molecules(PRMs). sClec play critical roles in the first line of immune defense. sClec has also shown that it is related to cell proliferation. The C-type lectins family in Periplaneta americana comprises a substantial number of members, with certain secretory subtypes exhibiting elevated expression during leg regeneration. Consequently, these lectins are hypothesized to be associated with the remarkable leg regeneration capacity of this species. We hope this clarifies the nature and significance of Clec genes. We have, accordingly, revised to include description and analysis of the Clec genes. [mention specific location: e.g., Page 7, Section 3.3, Lines 253-259; Page 10, Section 3.5, Lines 335-343]
Comments 4:In Figure 7, Did was evaluated by qPCR or Digital PCR? It is not clear in lines 276-277 what it means.
Response 4: Thank you for pointing this out. We appreciate the opportunity to clarify this point. The date presented in Figure 7 were generated using Digital PCR and the results of qRT-PCR. We have revised the manuscript text and Figure 7 to make our purpose clearer.[mention specific location: e.g., Page 10, Section 3.5, Lines 321-323; Page 11, Section 3.5, Figure 7]
Comments 5:Figure 8C-E, the Figure legend is not clear.
Response 5: Thank you for pointing this out. We have modified these figures by changing the scale bar from 50 mm to 100mm and increasing the font size of the scale bar label. The revised figures have been updated accordingly.[mention specific location: e.g., Page 12, Section 3.6, Figure 8C-E]
Comments 6: All the figures refer to elements "as indicated by the arrows," yet no arrows are actually present in the images. Additionally, the figure legends are poorly described and lack sufficient detail. They should be revised to include clear, accurate, and informative explanations that help the reader interpret the figures effectively.
Response 6: Thank you for pointing this out. We agree with this comment. Therefore, we have carefully revised the figures and their corresponding legends. We have added concept maps to help the reader interpret the figures effectively.[mention specific location: e.g., Page 5, Section 3.1, Lines 184-188, Figure 1]
Comments 7: Tukey's test is not the correct test to compare two groups.
Response 7: Thank you for your careful review and valuable feedback. The reviewer is correct. The statement indicating the use of Tukey's test for comparing the two groups was an error in our manuscript description. Tukey's test is indeed designed for multiple comparisons after an ANOVA, not for direct two-group comparisons. In our actual data analysis, we used an independent samples t-test to compare the means between the two groups. We sincerely apologize for this oversight in reporting the statistical method. We will correct this to accurately reflect the use of the t-test in the revised manuscript.[mention specific location: e.g., Page 13, Section 3.6, Lines 393-394, Figure 8]
Comments 8: Why did the authors compare the 0 vs 7 and 7-14? Why didn´t they compare 0 vs 7, and 0 vs 14 dpa, since 0 is the reference?
Response 8: We Thank the reviewer for this comment. The original comparisons(0 dpa vs 7 dpa, 7 dpa vs 14 dpa) were designed to highlight stage-specific transitions during leg regeneration.However, we fully agree that comparing each time point directly to the baseline(0 dpa) is essential for contextualizing overall changes relative to the starting point. We have included heatmap and the explicit comparison between 0 dpa and 14 dpa.[mention specific location: e.g., Page 7, Section 3.3, Lines 222-229, Figure 4 and Figure 5; Page 9, Section 3.4, Lines 297-299]
Comments 9: Results have to explain why and how they did the experiments.
Response 9: Thank you for pointing this out. We agree with this comment. Therefore, we have modified the results section to explain experiment’s purpose and content.[mention specific location: e.g., Page 10, Section 3.5, Lines 321-323; Page 11, Section 3.6, Lines 352-368]
Comments 10: Discussion has to support these results with literature.
Response10: As suggested by the reviewer, we have added more references to support our results.[mention specific location: e.g., Page 7, Section 3.3, Lines 238-270; Page 10, Section 3.5, Lines 335-343; Page 11, Section 3.6, Lines 355-363; Page 12, Section 3.6, Lines 381]
Comments 11: The conclusion is a summary; there is no conclusion in the current work.
Response11: Thank you for pointing this out. We have revised the conclusions and enhanced the interpretation of findings, limitations and implications.[mention specific location: e.g., Page 13, Section 4, Lines 396-405]
Comments 12: The manuscript needs to be reviewed by a native English editor. It contains several mistakes, for example: Lines 276–277: It says "of the digital expression date." Should this be "data" instead?
Response 12: Thank you for your valuable feedback. As suggested, we have engaged a professional native English-speaking editor to thoroughly revise the manuscript. We believe the language quality has been significantly improved. We have revised this description to enhance understanding.[mention specific location: e.g., Page10, Section 3.5, Lines 321-323]
Comments 13: Some information in the supplementary data is written in Chinese.
Response 13: Thanks for your careful checks. We are sorry for our carelessness. Based on your comments, we have made the corrections.
Reviewer 2 Report
Comments and Suggestions for Authors
This study by Liu et al. explores the molecular basis of limb regeneration in the American cockroach (Periplaneta americana) through comparative transcriptomic analysis. By analyzing differentially expressed genes (DEGs) during key time points in leg regeneration and validating candidates with RNA interference (RNAi), the authors identify EGFR signaling and C-type lectins as important factors involved in regenerative processes. The study’s strength lies in its combination of transcriptomics and functional validation, offering valuable insights into regeneration in hemimetabolous insects.
Major points:
- The study analyzes three timepoints following injury: immediately after amputation (0 dpa), 7 days and 14 days post-amputation. Yet comparisons are always made pairwise. A comparison of 0 and 14 dpa samples would be an important resource. For example, the authors refer to EGFR being downregulated in 7 dpa (0 vs 7 dpa) but then upregulated in 14 dpa (7 vs 14 dpa). However, for the reader, an important piece of information would be to understand whether the gene’s expression level is back up to the original baseline levels or even more upregulated than compared to 0 dpa. Additionally, representation of such data in heatmap form would be more digestible (instead of just Table 1).
- A better justification of why these genes were selected for further analysis is required. It is not immediately clear why the authors went after growth-related genes, as these do not need to be regeneration-specific, but rather could be related to cell-cycle regulation that could be responsible for normal wear-and-tear. Expanding the rationale in this area is important.
- Since the RNA-seq and qPCR are performed on bulk tissue, to support any claims suggesting functional EGFR and Regenectin interaction, the authors need to establish where these genes are expressed in the regenerating leg. In situ hybridization or antibody-based assays are required.
- Suggesting that Regenectin may act as a growth factor without protein-level evidence or receptor binding studies is an overreach. These statements should be toned down.
- The conclusion section needs some work. The authors need to expand on their interpretation of their findings, limitations, and future implications.
Minor points:
- The paper needs a clearer articulation of the central hypothesis in the introduction and discussion. Currently, the narrative bounces between exploratory transcriptomics and specific hypotheses about EGFR and Regenectin.
- Figure 1 needs to be improved. The legend needs to be expanded to reference all panels. The timepoint reflected in each panel should be presented on the figure, along with labeling of the key anatomical structures (the amputation plane, stump, wound epidermis and blastema). It's unclear how many biological replicates were used for histology, and whether variation in regenerative outcomes was observed. These details should be added.
- Figure 1A is not referenced in the text
- Figure 7 could benefit from the inclusion of all timepoints sampled in the RNA-seq. This would give the reader an overview of the broad regulation of the gene throughout the regenerative process.
- The conclusion/discussion should include a sentence about the limitation of analyzing transcriptional levels of receptors, as these do not inform us about the activity levels of the receptors/pathways.
- A single citation convention needs to be adopted (in line with the journal’s guidelines). A couple of inconsistencies are noted here:
- Line 44: citation after punctuation, but others before punctuation
- Line 134: citation has a space to separate it from the word preceding it, but others don’t.
- Line 37-38: Italicize the species Mus musculus.
The overall English is understandable, but has multiple minor errors, which should be corrected in a revision. A professional tone needs to be adopted, avoiding verbiage such as “and so on” and “besides”. Examples needing attention:
-
- Line 22: Regenectin may regulates leg regeneration
- Line 205: Some growth-related genes, such as EGFR, pointed, G1/S-specific cyclin-D2 and collagen alpha-1(II) chain, were significantly differentially (sentence ends abruptly)
- Line 231-232: Upregulated and Downregulated genes (U and D here should not be capitalized)
Author Response
Comments 1: The study analyzes three timepoints following injury: immediately after amputation (0 dpa), 7 days and 14 days post-amputation. Yet comparisons are always made pairwise. A comparison of 0 and 14 dpa samples would be an important resource. For example, the authors refer to EGFR being downregulated in 7 dpa (0 vs 7 dpa) but then upregulated in 14 dpa (7 vs 14 dpa). However, for the reader, an important piece of information would be to understand whether the gene’s expression level is back up to the original baseline levels or even more upregulated than compared to 0 dpa. Additionally, representation of such data in heatmap form would be more digestible (instead of just Table 1).
Response 1: Thank you for point this out. The original comparisons(0 dpa vs 7 dpa, 7 dpa vs 14 dpa) were designed to highlight stage-specific transitions during leg regeneration.However, we fully agree that comparing each time point directly to the baseline(0 dpa) is essential for contextualizing overall changes relative to the starting point. As suggested by the reviewer, we visualized the expression levels of these genes across various stages using heatmap.[mention specific location: e.g., Page 7, Section 3.3, Lines 222-229, Figure 4 and Figure 5; Page 9, Section 3.4, Lines 297-299]
Comments 2: A better justification of why these genes were selected for further analysis is required. It is not immediately clear why the authors went after growth-related genes, as these do not need to be regeneration-specific, but rather could be related to cell-cycle regulation that could be responsible for normal wear-and-tear. Expanding the rationale in this area is important.
Response 2: Thank you for pointing this out. We agree with this comment. The second Keystone symposium expended conceptual frameworks for regeneration by emphasizing growth mechanisms and it is necessary to understand the relationship between regenerative growth versus homeostatic growth control/matintrnance. Therefore, we have expand the rationale in this area.[mention specific location: e.g., Page 2, Section 1, Lines 78-81; Page 6, Section 3.3, Lines 210-214; Page 7, Section 3.3, Lines 238-270; Page 10, Section 3.5, Lines 335-343; Page 11, Section 3.6, Lines 351-368]
Comments 3: Since the RNA-seq and qPCR are performed on bulk tissue, to support any claims suggesting functional EGFR and Regenectin interaction, the authors need to establish where these genes are expressed in the regenerating leg. In situ hybridization or antibody-based assays are required.
Response 3: We agree that the in site hybridization or antibody-based assays would be useful to understand the details of their interaction. At this point, there has been a publication by Toshimitsu Arai showing that Regenectin mRNA expression exclusively within the newly formed epidermis of regenerating limbs[mention specific location: e.g., Page 12, Section 3.6, Lines 360-363, Ref. 65]. We have added this citation and reference to the manuscript. We have tried to verify the expression location of EGFR protein by antibodies of other species. However, no fluorescence signal has been detected in immunofluorescence. We hope, in the future, to employ In site hybridization to determine the expreesion location of EGFR gene and Co-ip to determine their interaction. This study focus on answering questions regarding the impact of growth- or regeneration-related genes on regeneration.
Comments 4: Suggesting that Regenectin may act as a growth factor without protein-level evidence or receptor binding studies is an overreach. These statements should be toned down.
Response 4: We agree with the reviewer’s assessment that suggesting Regenectin acts directly as a growth factor without protein-level evidence or receptor binding studies was overstated. Accordingly, throughout the manuscript, we have revised the manuscript to remove these overstated statements.[mention specific location: e.g., Page 2, Section 1, Line 81; Page 10, Section 3.5, Line 343]
Comments 5: The conclusion section needs some work. The authors need to expand on their interpretation of their findings, limitations, and future implications.
Response 5: Thank you for pointing this out. We have revised the conclusions and enhanced the interpretation of findings, limitations and implications.[mention specific location: e.g., Page 13, Section 4, Lines 396-405]
Comments 6: The paper needs a clearer articulation of the central hypothesis in the introduction and discussion. Currently, the narrative bounces between exploratory transcriptomics and specific hypotheses about EGFR and Regenectin.
Response 6: Think you for pointing this out. We agree with this comment. Therefore, we have, accordingly,modified our introduction and discussion to emphasize this point.[mention specific location: e.g., Page 2, Section 1, Lines 73-81; Page 6, Section 3.3, Lines 210-214; Page 7, Section 3.3, Lines 238-270; Page 10, Section 3.5, Lines 335-343; Page 11, Section 3.6, Lines 352-368]
Comments 7: Figure 1 needs to be improved. The legend needs to be expanded to reference all panels. The timepoint reflected in each panel should be presented on the figure, along with labeling of the key anatomical structures (the amputation plane, stump, wound epidermis and blastema). It's unclear how many biological replicates were used for histology, and whether variation in regenerative outcomes was observed. These details should be added.
Response 7: We sincerely thank the reviewer for these valuable suggestions to improve the completeness of Figure 1. We have carefully revised the figure and its legend accordingly. The legend for Figure 1 has been comprehensively expended to describe the content(A-E). The specific timepoint is now clearly indicated directly on each corresponding panel within the figure and legend. The anatomical structures relevant to leg regeneration are circled with green dotted lines.[mention specific location: e.g., Page 3, Section 2.2, Lines 97; Page 5, Section 3.1, Figure 1, Lines 184-188]
Comments 8: Figure 1A is not referenced in the text
Response 8: Thank you for pointing this out. We acknowledge the point that Figure 1A was not referenced in the text. After modifying Figure 1, we have added a specific reference to original Figure 1A in the revised manuscript.[mention specific location: e.g., Page 5, Section 3.1, Lines 169]
Comments 9: Figure 7 could benefit from the inclusion of all timepoints sampled in the RNA-seq. This would give the reader an overview of the broad regulation of the gene throughout the regenerative process.
Response 9: Your suggestion really means a lot to us. Yes, it would be more understandable if we showed the inclusion of all timepoints sampled in the RNA-seq. Therefore, we have modified Figure 7 to give the reader an overview of the broad regulation of the gene throughout the regenerative process.[mention specific location: e.g., Page 11, Section 3.5, Figure 7]
Comments 10: The conclusion/discussion should include a sentence about the limitation of analyzing transcriptional levels of receptors, as these do not inform us about the activity levels of the receptors/pathways.
Response 10: We think this is an excellent suggestion. We have explicitly acknowledged this limitation in the revised Discussion section and Conclusion section. This addition clarifies that while our study provides insights into gene expression regulation, it does not assess the actual activation or signaling flux through these receptor pathways.[mention specific location: e.g., Page 12, Section 3.6, Lines 384-388; Page 13, Section 4, Lines 402-405]
Comments 11: A single citation convention needs to be adopted (in line with the journal’s guidelines). A couple of inconsistencies are noted here:
Line 44: citation after punctuation, but others before punctuation
Line 134: citation has a space to separate it from the word preceding it, but others don’t.
Line 37-38: Italicize the species Mus musculus.
Response 11: We were really sorry for our careless mistakes. Thanks for your reminder. We have carefully checked the manuscript and corrected the errors accordingly.[mention specific location: e.g., Page 1, Section 1, Line 41; Page 2, Section 1, Lines 47; Page 4, Section 2.5, Line 135]
Comments 12: Comments on the Quality of English Language
The overall English is understandable, but has multiple minor errors, which should be corrected in a revision. A professional tone needs to be adopted, avoiding verbiage such as “and so on” and “besides”. Examples needing attention:
Line 22: Regenectin may regulates leg regeneration
Line 205: Some growth-related genes, such as EGFR, pointed, G1/S-specific cyclin-D2 and collagen alpha-1(II) chain, were significantly differentially (sentence ends abruptly)
Line 231-232: Upregulated and Downregulated genes (U and D here should not be capitalized)
Response 12: Thank you for your valuable feedback. As suggested, we have engaged a professional native English-speaking editor to thoroughly revise the manuscript. We believe the language quality has been significantly improved. Thanks for your careful checks.We feel sorry for our carelessness. In our resubmitted manuscript, the typo is revised. Thanks for your correction.[mention specific location: e.g., Page 1, Abstract, Line 27; Page 6, Section 3.3, Lines 210-235; Page 9, Section 3.4, Lines 283-299]
Round 2
Reviewer 1 Report
Comments and Suggestions for Authors
The authors only addressed the concern related to the arrows in Figure 1. They corrected the English; however, the manuscript still has the same issues as the beginning. No improvement in the content, discussion, and methods was made. In the response, they affirm they did make changes, but they didn´t. Paraphrasing the manuscript only improves the reading, but not the scientific content. The manuscript still has major flaws.
Author Response
Comments 1: In the introduction, the authors mention traditional medicine. However, there is no scientific evidence supporting the claim that raw or processed cockroach products promote regeneration. The manuscript appears to promote this unverified concept, which borders on pseudoscience and should be critically addressed or removed.
Response 1: We agree with the reviewer’s assessment. We have deleted this description since this claim lacked supporting evidence and could mislead readers, it was removed from the manuscript.[mention specific location: e.g., Page 2, Section 1, Line 81]
Comments 2:The manuscript includes a meta-analysis using several tools such as KEGG, GO, and others. However, the rationale for using these tools is not provided, nor are any of them explained. Despite performing such a comprehensive analysis, the authors only highlight 22 "interesting" genes without discussing the remaining data. Given that the transcriptome includes over 50,000 genes. The authors need to provide a more thorough analysis and discussion of the broader findings.
Response 2: Thank you for pointing this out. we agree with this comment. Therefore, we have explained the principles underlying these tools. The rationale for using these tools is to understand the functional implications of the identified DEGs. The functions of them were explained, such as the GO database provides functional annotation of screened genes, encompassing biological processes (BP), cellular components (CC), and molecular functions (MF) and the KEGG database categorizes genes within a gene set based on their associated biochemical pathways or molecular functions. [mention specific location: e.g., Page 4, Section 2.6, Lines 145-150] Although the transcriptome contains over 50,000 genes, only approximately 3,000 are differentially expressed. Furthermore, the functions of many genes remain uncharacterized, thus limiting comprehensive functional analyses. Therefore, this study focused on screening DEGs associated with growth or regeneration in Periplaneta americana. We experimentally validated their effects on regeneration in P. americana, establishing a theoretical framework for investigating its regeneration mechanisms. Specifically, in the Results and Discussion, we conducted a broader screening of growth- or regeneration-related differentially expressed genes and visualized the expression levels of these genes across various stages using heatmap.[mention specific location: e.g., Page 6, Section 3.3, Lines 216-277, Figure 5] Additionally, we have modified Figure 7 to give the reader an overview of the broad regulation of the gene throughout the regenerative process.[mention specific location: e.g., Page 11, Section 3.5, Figure 7]
Comments 3:What are Clec genes, and why are they important?
Response 3: We thank the reviewer for this important question regarding Clec genes. Clec genes encode a large family of proteins called C-type lectins or CTLs. sClec genes encode a distinct subgroup of soluble C-type lectins. Unlike the membrane bound Clec receptors, these proteins are secreted into body fluids where they function as pattern recognition molecules(PRMs). sClec play critical roles in the first line of immune defense. sClec has also shown that it is related to cell proliferation. The C-type lectins family in Periplaneta americana comprises a substantial number of members, with certain secretory subtypes exhibiting elevated expression during leg regeneration. The upregulated expression of C-type lectin during regeneration suggests their significant contribution to leg regeneration in Periplaneta americana. Consequently, we hope this clarifies the nature and significance of Clec genes. We have, accordingly, added to include description and analysis of the Clec genes. [mention specific location: e.g., Page 7, Section 3.3, Lines 261-267; Page 10, Section 3.5, Lines 343-351]
Comments 4:In Figure 7, Did was evaluated by qPCR or Digital PCR? It is not clear in lines 276-277 what it means.
Response 4: Thank you for pointing this out. We appreciate the opportunity to clarify this point. The date presented in Figure 7 were generated using Digital PCR and the results of qRT-PCR. We have revised the manuscript text and Figure 7 to make our purpose clearer.[mention specific location: e.g., Page 10, Section 3.5, Lines 329-331; Page 11, Section 3.5, Figure 7A-J]
Comments 5:Figure 8C-E, the Figure legend is not clear.
Response 5: Thank you for pointing this out. We have modified these figures by changing the scale bar from 50 mm to 10mm and increasing the font size of the scale bar label. The revised figures have been updated accordingly.[mention specific location: e.g., Page 13, Section 3.6, Figure 8C-E]
Comments 6: All the figures refer to elements "as indicated by the arrows," yet no arrows are actually present in the images. Additionally, the figure legends are poorly described and lack sufficient detail. They should be revised to include clear, accurate, and informative explanations that help the reader interpret the figures effectively.
Response 6: Thank you for pointing this out. We agree with this comment. Therefore, we have carefully revised the figures and their corresponding legends. We have added concept maps to help the reader interpret the figures effectively.[mention specific location: e.g., Page 5, Section 3.1, Lines 189-193, Figure 1]
Comments 7: Tukey's test is not the correct test to compare two groups.
Response 7: Thank you for your careful review and valuable feedback. The reviewer is correct. The statement indicating the use of Tukey's test for comparing the two groups was an error in our manuscript description. Tukey's test is indeed designed for multiple comparisons after an ANOVA, not for direct two-group comparisons. In our actual data analysis, we used an independent samples t-test to compare the means between the two groups. We sincerely apologize for this oversight in reporting the statistical method. We will correct this to accurately reflect the use of the t-test in the revised manuscript.[mention specific location: e.g., Page 13, Section 3.6, Lines 403-405, Figure 8]
Comments 8: Why did the authors compare the 0 vs 7 and 7-14? Why didn´t they compare 0 vs 7, and 0 vs 14 dpa, since 0 is the reference?
Response 8: We Thank the reviewer for this comment. The original comparisons(0 dpa vs 7 dpa, 7 dpa vs 14 dpa) were designed to highlight stage-specific transitions during leg regeneration.However, we fully agree that comparing each time point directly to the baseline(0 dpa) is essential for contextualizing overall changes relative to the starting point. We have included heatmap and the explicit comparison between 0 dpa and 14 dpa.[mention specific location: e.g., Page 7, Section 3.3, Lines 228-236, Figure 4 and Figure 5; Page 9, Section 3.4, Lines 297-308]
Comments 9: Results have to explain why and how they did the experiments.
Response 9: Thank you for pointing this out. We agree with this comment. Therefore, we have modified the results section to explain experiment’s purpose and content.[mention specific location: e.g., Page 4, Section 3.1, Lines 173-175; Page 10, Section 3.5, Lines 321-323; Page 11, Section 3.6, Lines 358-365; Page 12, Section 3.6, Lines 366-372]
Comments 10: Discussion has to support these results with literature.
Response10: As suggested by the reviewer, we have added more references to support our results.[mention specific location: e.g., Page 7, Section 3.3, Lines 245-277; Page 10, Section 3.5, Lines 343-351, References 35-40; Page 11, Section 3.6, Lines 358-369, References 67-69; Page 12, Section 3.6, Line 390, Reference 70]
Comments 11: The conclusion is a summary; there is no conclusion in the current work.
Response11: Thank you for pointing this out. We have revised the conclusions and enhanced the interpretation of findings, limitations and implications.(This study investigated the morphological changes during leg regeneration of Periplaneta americana. Through comparative transcriptome analysis, we identified 86 DEGs associated with growth or regeneration processes, providing valuable data for future research into the molecular mechanisms of leg regeneration in P. americana. The findings demonstrate that epidermal growth factor receptor (EGFR) and secreted C-type lectin (Regenectin) regulate leg regeneration. Additionally, Regenectin modulates the expression of EGFR and its ligand Spitz. The dynamic expression patterns and regulatory interactions between EGFR and Regenectin during leg regeneration warrant further investigation. These findings advance our understanding of the molecular mechanisms underlying leg regeneration in P. americana.)[mention specific location: e.g., Page 13, Section 4, Lines 409-418]
Comments 12: The manuscript needs to be reviewed by a native English editor. It contains several mistakes, for example: Lines 276–277: It says "of the digital expression date." Should this be "data" instead?
Response 12: Thank you for your valuable feedback. As suggested, we have engaged a professional native English-speaking editor to thoroughly revise the manuscript. We believe the language quality has been significantly improved. We have revised this description to enhance understanding.(To experimentally validate the transcriptome sequencing results, ten growth-related DEGs were selected from the coxa of P. americana for qPCR analysis.)[mention specific location: e.g., Page10, Section 3.5, Lines 329-331]
Reviewer 2 Report
Comments and Suggestions for Authors
The authors have substantially improved the manuscript in response to reviewer feedback. Key revisions include clearer articulation of the central hypothesis, expanded justification for gene selection, enhanced data visualization (including heatmaps and full timepoint comparisons), and improved figure quality and legends. Overinterpretations regarding Regenectin’s function were appropriately tempered, and limitations around receptor expression and activity were acknowledged. While spatial validation of EGFR expression remains incomplete, this limitation is now transparently discussed. Minor revisions are still recommended to specify the number of biological replicates that were used for Fig 8 C-E in the legends so the reader can appreciate how often these phenotypes are observed. With these final adjustments, the manuscript will be ready for publication.
Author Response
Comments 1: Minor revisions are still recommended to specify the number of biological replicates that were used for Fig 8 C-E in the legends so the reader can appreciate how often these phenotypes are observed.
Response 1: We think this is an excellent suggestion. We have added the Fig. 8F to depict the limb regeneration phenotype, quantified using the relative regeneration length index. We believed that Fig. 8F can make the reader appreciate how often these phenotypes are observed.[mention specific location: e.g., Page 13, Section 3.6, Lines 384-385 and 400-407, Fig.8F]